# OPTIMIZING SUCCESS RATE IN REINFORCEMENT LEARNING VIA LOOP PENALTY

## ABSTRACT

Current reinforcement learning generally uses discounted return as its learning objective. However, real-world tasks may often demand a high success rate, which can be quite different from optimizing rewards. In this paper, we explicitly formulate the success rate as an *undiscounted* form of return with $\{0, 1\}$-*binary* reward function. Unfortunately, applying traditional Bellman updates to value function learning can be problematic for learning undiscounted return, and thus not suitable for optimizing success rate. From our theoretical analysis, we discover that values across different states tend to converge to the same value, resulting in the agent wandering around those states without making any actual progress. This further leads to reduced learning efficiency and inability to complete a task in time. To combat the aforementioned issue, we propose a new method, which introduces Loop Penalty (LP) into value function learning, to penalize disoriented cycling behaviors in agent's decision-making. We demonstrate the effectiveness of our proposed LP on three environments, including grid-world cliff-walking, Doom first-person navigation and robot arm control, and compare our method with Q-learning, Monte-Carlo and Proximal Policy Optimization (PPO). Empirically, LP improves the convergence of training and achieves a higher success rate.

## 1 INTRODUCTION

Reinforcement learning usually adopts expected discounted return as objective, and has been applied in many tasks to find the best solution, e.g. finding the shortest path and achieving the highest score (Sutton & Barto, 2018; Mnih et al., 2015; Shao et al., 2018). However, many real-world tasks, such as robot control or autonomous driving, may demand more in success rate (i.e. the probability for the agent to fulfill task requirements) since failures in these tasks may cause severe damage or consequences. Previous works commonly treat optimizing rewards equivalent to maximizing success rate (Zhu et al., 2018; Peng et al., 2018; Kalashnikov et al., 2018), but their results can be error-prone when applied to real-world applications.

We believe that success rate is different from expected discounted return. The reasons are as follows: 1) expected discounted return commonly provides dense reward signals for transitions in an episode, while success or not is a *sparse* binary signal only obtained at the end of an episode; 2) expected discounted return commonly weights results in the immediate future more than potential rewards in the distant future, whereas success or not does not have such a weighting and is only concerned about the overall or the final result. Policies with high expected discounted returns are often more demanding in short-term performance than those with high success rates and optimizing success rates often leads to multiple solutions. As a result, policies with high success rates tend to be more reliable and risk-averse while policies with high expected discounted returns tend to be risk-seeking.

See the cliff-walking example in Fig. 1 where the objective is to walk from the origin state marked with a triangle to the destination state marked with a circle. The "Slip" area in light grey winds with a certain probability $p_{\text{fall}} = 0.1$, making the agent uncontrollably move down; the dark gray area at the bottom row denotes "Cliff". In Fig. 1, the blue trajectory shown on the left is shorter but riskier than the green one shown on the right. In commonly-used hyperparameter settings, such as $\gamma = 0.9$, the agent tends to follow the blue trajectory rather than the green one, although the green trajectory has a higher success rate.

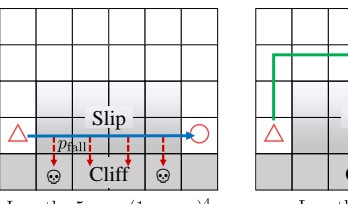 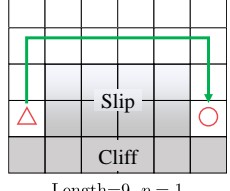

Length=5, $p = (1 - p_{\text{fall}})^4$     Length=9, $p = 1$

Figure 1: Cliff-walking example

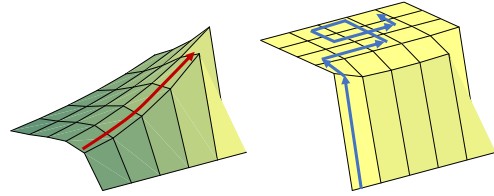

Figure 2: Illustration of value fucntion discounted (left) and undiscounted (right)

We acknowledge that for this simple example, optimizing expected discounted return with a careful design of $\gamma$ that meets $(1 - p_{\text{fall}})^4 < \gamma^{9-5}$ can produce a policy with the highest success rate. However, this result relies on task-specific knowledge about the environment, generally not available in more complex tasks. These findings lead us to the following question: can we express success rate in a general form so that it can be directly optimized? In this paper, we discover a universal way of representing success rate is to 1) use a $\{0, 1\}$-binary reward indicates whether or not a trajectory is successful, and 2) set $\gamma = 1$ so that the binary signal back-propagates without any discount.

Unfortunately, this expression belongs to undiscounted problems and the convergence of value iteration often cannot be guaranteed (Xu et al., 2018). Nevertheless, we can still explicitly solve the Bellman equation in a matrix form for the special undiscounted return (success rate). We derive that if the transition dynamics of the environment permit existence of an irreducible ergodic set of states, $\gamma = 1$ will lead to an undesirable situation: state or state-action values tend to converge to the same value, which we refer to as *uniformity*. As shown in Fig. 2 for the contour of state values in our cliff-walking example, uniformity is reflected as a plateau in the right figure, which is caused by non-discounting and does not exist in discounting cases (left figure). Uniformity makes the selection of actions purposeless within the plateau, resulting in disoriented and time-consuming behaviors in the agent's decision-making, and unsatisfactory success rates.

Based on the above analysis, we introduce Loop-Penalty (LP) into value function learning to penalize disoriented and cycling behaviors in trajectories. We derive that this penalty can be realized by multiplying a special mask function to the original value function. Note that our strategy is general and is applicable to many RL algorithms. We provide concrete loss functions for three popular algorithms in this paper: Monte Carlo, Deep Q-learning and Proximal Policy Optimization (Schulman et al., 2017). We verify the effectiveness in three representative environments: grid-world cliff-walking, vision-based robot grasping, and first-person navigation in 3D Vizdoom (Kempka et al., 2016), showing that LP can alleviate the uniformity problem and achieve better performance. Finally, we summarize the major contributions of our paper in the following:

- We formally introduce the objective of "success rate" in reinforcement learning. Our formulation of success rate is general and is applicable for many different RL tasks.

- We theoretically analyze the difficulty in optimizing success rate and show that the uniformity among state values and the resulting loops in trajectories are the key challenges.

- We propose LP which can be combined with any general RL algorithm. We demonstrate empirically that LP can alleviate the problem of "uniformity" among state values and significantly improve success rates in both discrete and continuous control tasks.

## 2 RELATED WORK

To the best of our knowledge, currently there is no research that adopts success rate directly as the learning objective. The reason is that success rate is usually not the main criterion in tasks investigated by RL, e.g. video games and simulated robot control. Although some studies used success rate to evaluate the performance of the policies (Andrychowicz et al., 2017; Tobin et al., 2018; Ghosh et al., 2018; Kalashnikov et al., 2018), they used task-specific reward design and discounted return during training, instead of directly optimizing success rate.

The notion of "success" may be reflected in constraints considered in the domain of safe RL (García & Fernández, 2015). Geibel & Wysotzki (2005) considered constraints on the agent's behavior and

discouraged the agent from moving to error states. Geibel (2006) studied constraints on the expected return to ensure acceptable performance. A. & Ghavamzadeh (2013) proposed constraints on the variance of some measurements to pursue an invariable performance. Previous studies have also considered safety in the exploration process (García & Fernández-Rebollo, 2012; Mannucci et al., 2018). Although these studies deemed success rate as an additional constraint in learning, they either simply assumed that the constraint can be certainly satisfied or penalized constraint violations.

The deficiency of expected discounted return as a training objective has been recognized by many studies. Instead of just optimizing expected return, Heger (1994); Tamar et al. (2013) adopted the minimax criterion that optimizes the worst possible values of the return. By doing so, occasional small returns would not be ignored at test time. Gilbert & Weng (2016); Chow et al. (2017) extended this idea to arbitrary quantiles of the return. However, all these studies are not optimizing success rate directly since they are based on a quantitative measurement of performance and are unnecessarily sensitive to the worst cases. In contrast, success rate is based on a binary signal which only distinguishes between success and failure.

Our work involves optimization of an undiscounted return. The instability in training towards an undiscounted return has been mentioned by Schwartz (1993); Xu et al. (2018). However, most studies on undiscounted return focused on continuous settings and considered the average reward as objectives (Schwartz, 1993; Ortner & Ryabko, 2012; Zahavy et al., 2020). There seems to be a general view that the instability in training towards undiscounted return only exists in continuous cases but not in episodic cases (Pitis, 2019). Contrary to this view, we propose that training instability also exists in episodic cases. For optimizing success rate, we provide a theoretical analysis and show the existence of training instability and propose a practical method that alleviates this problem.

## 3 SUCCESS RATE IN REINFORCEMENT LEARNING

In this section we provide a formal definition of success rate, explain its relationship with expected discounted sum of rewards, and analyze the problems in optimizing success rate.

### 3.1 SUCCESS RATE

In RL, given a policy $\pi$, success rate specifically refers to the ratio of the successful trajectories to all trajectories. As in a general setting of RL, a trajectory is expressed as $\tau = \{(s_0, a_0, r_0), \ldots, (s_T, a_T, r_T), s_{T+1}\}$ rolled out by following policy $\pi$, where $s_t \in \mathcal{S}$ is state, $a_t \in \mathcal{A}$ denotes action, $r_t$ represents immediate reward and $T$ is the length of the trajectory. Because the notion of success should only depend on the visited states in a trajectory, we concisely express "success" by defining a set of desired states $\mathcal{S}_g \subset \mathcal{S}$ that denote task completion, e.g. the destination state in our cliff-walking example. At a high level, the goal of the agent is to reach any state in $\mathcal{S}_g$ within a given planning horizon $T$, and the environment terminates either upon arriving at a desired state or reaching a maximum allocated timestep $T$. Without loss of generality, we say that "a trajectory $\tau$ is successful" if and only if $\tau_{-1} \in S_g$, where $\tau_{-1}$ is the last state in $\tau$. Formally, we use an indicator function $I(s \in S_g)$ to denote success, where $I(\cdot)$ takes value of 1 when the input statement is true and 0 otherwise. Since this expression is task-independent, our analysis can be widely applicable. Accordingly, we formally define the success rate as follows:

**Definition 1.** The *success rate* of a given policy $\pi$ is defined as

$$\beta_\pi(s_0) = \sum_\tau p_\pi(\tau|s_0) I(\tau_{-1} \in S_g) \tag{1}$$

where $p_\pi(\tau|s_0) = \prod_{t=0}^{T} \pi(a_t|s_t) p(s_{t+1}|s_t, a_t)$ is the probability of observing trajectory $\tau$.

In order to find a policy that optimizes success rate, we derive a recursive form of policy evaluation similar to the Bellman equation (Sutton & Barto, 2018), as shown in Theorem 1.

**Theorem 1.** The success rate is a state-value function represented as an expected sum of undiscounted return, with the reward function $R(s)$ defined to take the value of 1 if $s \in S_g$, 0 otherwise.

*Proof sketch*: We segment the trajectories and generate sub-trajectories, $\tau \in \Gamma, \tau_{0:k} \in \hat{\Gamma}$, where $k \in (0, T]$. Note that $\Gamma = \hat{\Gamma}$, because 1) $\forall \tau \in \Gamma$, we have $\tau_{0:T} \in \hat{\Gamma} = \tau, \Gamma \subseteq \hat{\Gamma}$, 2) $\tau_{0:k}$

is a trajectory, $\hat{\Gamma} \subseteq \Gamma$. Then the success rate $\beta_\pi(s_t)$ can be rewritten as the product sum of the probability of reaching $s_{t+k}$ and the indicator $I(\tau_{s_{t+k}} \in S_g)$ for all $s_{t+k}$:

$$\beta_\pi(s_t) = \sum_{k=1}^{T-t} \sum_{s_{t+k}} p_\pi(s_{t+k}|s_t) I(s_{t+k} \in S_g) \tag{2}$$

where $p_\pi(s_{t+k}|s_t)$ the probability of reaching $s_{t+k}$ from $s_t$. Complete proof is in appendix. $\square$

Therefore, we can optimize success rate through setting the above $\{0, 1\}$-binary reward function and adopting an undiscounted form of return. The problem is that this formulation falls into optimizing the undiscounted form of return and may have problems in training stability (Xu et al., 2018).

### 3.2  Uniformity in success rate optimization

In the following part, we will show that $\gamma = 1$ can cause uniformity among state values, resulting in possible loops in trajectories, which hurts training stability.

*A. The concept of uniformity*

First, we define the concept of *uniformity*. Given a policy $\pi$, we say that *uniformity* arises when the state-value estimates of a set of strongly connected states become the same. Here we say two states are strongly connected if one state is reachable from the other and vice versa, e.g. the first two rows in the grid-world example (Fig. 1). Since state value represents the expected sum of available rewards (Sutton & Barto, 2018), uniformity means that moving in this connected area/region will potentially lead to the same amount of return. This phenomenon can hardly occur with discounted return since the discounting poses a preference for time-efficiency in collecting rewards and penalizes purposeless wandering. However, uniformity may happen when the objective is success rate since efficient trajectories and inefficient ones become indistinguishable.

*B. Proof of the existence of uniformity*

In this section, we theoretically prove that $\gamma = 1$ in the expression of success rate can cause uniformity. Because uniformity is a phenomenon about concrete state values, common techniques used to analyze the overall performance such as regret bound and contraction mapping do not apply here. Hence, we directly solve the Bellman equation to get state values. As for the reward function, we are fortunate that in our case the reward function only takes $\{0, 1\}$-binary values, which makes our analysis tractable. As for the optimization process, we analyze state values at convergence by first assuming a policy with uniformity, and then show that this policy will be kept during optimization.

For succinctness in description, we assume $\mathcal{S}$ to be finite to write the Bellman equation into a matrix form: $V = \boldsymbol{P}^\pi R + \gamma \boldsymbol{P}^\pi V$, where $V, R \in \mathbb{R}^{|\mathcal{S}|}$, $\boldsymbol{P}^\pi \in \mathbb{R}^{|\mathcal{S}| \times |\mathcal{S}|}$ and $|\mathcal{S}|$ is the cardinality of the state space. Without loss of generality, we denote the desired states at the bottom of each vector, so $R = [0, \ldots, 0, 1, \ldots, 1]^\mathrm{T}$. Then we formulate the concept of "area" as a set of states $\mathcal{S}_e \subsetneq \mathcal{S}$ that are irreducible ergodic in the Markov process conditioned on a policy $\pi$. By assuming the existence of $\pi$ and $\mathcal{S}_e$, and denoting states in $\mathcal{S}_e$ as the first $|\mathcal{S}_e|$ elements in the vectors, the $\pi$-conditioned transition probability matrix can be divided into

$$\boldsymbol{P}^\pi = \begin{bmatrix} \boldsymbol{P}_{ee}^\pi & \boldsymbol{O} \\ \boldsymbol{P}_{oe}^\pi & \boldsymbol{P}_{oo}^\pi \end{bmatrix}, \tag{3}$$

where $\boldsymbol{P}_{ee}^\pi$ is the transition probability matrix for $s \in \mathcal{S}_e$. Accordingly, we have the following Bellman equation for $s \in \mathcal{S}_e$:

$$V_e = \boldsymbol{P}_{ee}^\pi R_e + \gamma \boldsymbol{P}_{ee}^\pi V_e = \gamma \boldsymbol{P}_{ee}^\pi V_e . \tag{4}$$

Analyzing uniformity requires solving Eq.4. For $\gamma < 1$, the solution is unique $V_e = [0, \ldots, 0]^\mathrm{T}$ because $\boldsymbol{P}_{ee}^\pi$ is a stochastic matrix and $(\boldsymbol{I} - \gamma \boldsymbol{P}_{ee}^\pi)$ must be non-singular, and the value $0$ drives the agent to leave $\mathcal{S}_e$ in future policy update. However, when $\gamma = 1$, there are infinite solutions, as established in the following theorem.

**Theorem 2.** For $\gamma = 1$, if $\mathcal{S}_e$ exists, the solution space of Eq.4 is $\{V_e = m \cdot [1, \ldots, 1]^\mathrm{T} | m \in \mathbb{R}\}$.

*Proof*: Because states in $\mathcal{S}_e$ are ergodic, for any start-distribution $u_1^\mathrm{T}$ and $u_2^\mathrm{T}$ among $\mathcal{S}_e$, we have

$$u_1^\mathrm{T} \lim_{i \to \infty} (\boldsymbol{P}_{ee}^\pi)^i = u_2^\mathrm{T} \lim_{i \to \infty} (\boldsymbol{P}_{ee}^\pi)^i . \tag{5}$$

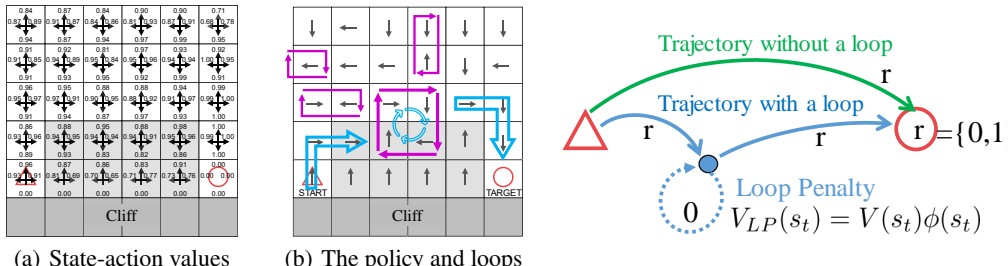

(a) State-action values      (b) The policy and loops

Figure 3: Numeric example of uniformity and loop     Figure 4: Sketch of Loop Penalty

Thus, $\lim_{i \to \infty}(\boldsymbol{P}_{ee}^{\pi})^i$ should be in the form that every row is the same, as illustrated below:

$$\lim_{i \to \infty}(\boldsymbol{P}_{ee}^{\pi})^i = \begin{bmatrix} x_1 & x_2 & \cdots & x_{|\mathcal{S}_e|} \\ x_1 & x_2 & \cdots & x_{|\mathcal{S}_e|} \\ \vdots & \vdots & \vdots & \vdots \\ x_1 & x_2 & \cdots & x_{|\mathcal{S}_e|} \end{bmatrix}$$

Note that all the elements are non-zero because $\mathcal{S}_e$ is irreducible. Thus, for equation $V_e = (\lim_{i \to \infty}(\boldsymbol{P}_{ee}^{\pi})^i)V_e$, the solutions are $m \cdot [1, \ldots, 1]^{\mathrm{T}}, m \in \mathbb{R}$. Because $\boldsymbol{P}_{ee}^{\pi}$ is a stochastic matrix, these solutions also satisfy Eq.4. Now, because solutions for Eq.4 also satisfy $V_e = (\lim_{i \to \infty}(\boldsymbol{P}_{ee}^{\pi})^i)V_e$, the solution spaces of the two equations become the same. Therefore, the solution space of Eq.4 is $\{V_e = m \cdot [1, \ldots, 1]^{\mathrm{T}} | m \in \mathbb{R}\}$, which completes the proof. □

This theorem demonstrates that when evaluating policy in terms of success rate, the converged values for states in $\mathcal{S}_e$ are the same and may take arbitrary values. This proves the existence of *uniformity* among state values.

Now we reason that there can be a policy $\pi$ that produces $\mathcal{S}_e$ and that this policy can be kept by the agent during policy optimization. (1) As for $\mathcal{S}_e$, it is common in RL environment that there is a set of two or more states that are reachable from each other without randomness. If the policy is initialized (or disturbed by random sampling during learning) to only stay in this set of states, then it gives the set of states $\mathcal{S}_e$. Note that the desired states are not in $\mathcal{S}_e$ because they are absorbing and cannot reach other states. This ensures that $R_e = [0, \ldots, 0]^{\mathrm{T}}$, by which Eq.4 is valid. (2) As for the agent keeping $\pi$ during policy optimization, we check if the state values satisfy the Bellman optimal equation. We have derived that any $m$ may be the value of state in $\mathcal{S}_e$. If the value $m$ is larger than the value of states reachable from $\mathcal{S}_e$ (probably due to initialization of value function), then the update target of values of states in $\mathcal{S}_e$ remains $m$. This means that $m$ satisfies the Bellman optimal equation at states in $\mathcal{S}_e$, and that the policy at $\mathcal{S}_e$ is kept during policy update. So far, we have proved that the objective of success rate can cause uniformity in state values.

*C. Problems caused by uniformity*

In RL, the agent selects actions based on the evaluation of future returns. When uniformity happens, the evaluation of different actions become the same, so the agent can only make random selections. This leads to disoriented, time-consuming but meaningless behaviors and an unsatisfactory success rate. In practice, because of disturbances due to random exploration, there may be slight differences between state values. Although this makes action-selection certain, it may result in undesirable policies, which causes instability in training. Fig. 3 shows a numeric example. We adopt Q-learning and illustrate the trained Q-values and the preferred actions respectively in (a) and (b). The Q-values are almost the same in upper grids, and there are several potential loops in the agent's trajectory. If the agent enters a loop, it will keep repeating the loop and fail in reaching the target.

## 4    METHOD: LOOP PENALTY

So far we have shown the problems in optimizing success rate. As for the solution, our insight is to suppress the generation of "loops" to penalize disoriented cycling behaviors in agents decision-making. In this section, we derive the cost function for minimizing the probability of loops, which

---

**Algorithm 1** Loop-Penalty Q-Learning

---

  **Initialize:** action-value function $\mathcal{Q}$, episode buffer $\mathcal{D}$;
  **for** episode$= 1, M$ **do**
    Initialise episode buffer $\mathcal{D}$;
    **for** $t = 1, T$ **do**
      With probability $\epsilon$ select a random action $a_t$, otherwise select $a_t = max_a Q(s_t, a)$;
      Execute action $a_t$ in emulator, get and store transition $(s_t, a_t, r_t, s_{t+1})$ in $\mathcal{D}$;
    **end for**
    **for** each transition $\{s_t, a_t, r_t, s_{t+1}\}$ in $\mathcal{D}$ **do**
      Initialise the marker factor of loop $\phi(s_t) \leftarrow 1$;
      **for** each $\{i, j | 0 < i < t, t < j < T\}$ **do**
        Calculate $\phi_t \leftarrow \phi_t \cap I(s_i \neq s_j)$;
      **end for**
      Set $y_t = \begin{cases} r_t & \text{for} \quad \text{terminal} \quad s_{t+1} \\ (r_t + max_a Q(s_{t+1}, a))\phi_t & \text{for} \quad \text{non-terminal} \quad s_{t+1} \end{cases}$;
      Perform a gradient descent step on $\|y_t - Q(s_t, a_t)\|_2$;
    **end for**
  **end for**

---

introduces Loop Penalty (LP) into value function learning. Then we introduce a practical algorithm that can implement this framework for reinforcement learning problems.

### 4.1 LOOP PENALTY

Our idea is that the agent not only needs to maximize the success rate, but also minimize the probability of "loops". This is formalized as follows:

$$\pi^* = \underset{\pi}{argmax} \; p_\pi(\tau^{no-loop}_{-1} \in S_g), \tag{6}$$

where $\tau^{no-loop}$ is the trajectory without loops where the agent visits some states more than once, in which the agent never revisits a previous state. We now derive the recursive state-value function $\beta^{loop-penalty}_\pi(s_t)$ with our loop-penalty for the optimization of Eq.6.

**Theorem 3.** The state-value function policy for Eq.6 is

$$\beta^{loop-penalty}_\pi(s_t) = \mathbb{E}_{\tau \sim \pi} \left[ I(s_{t+1} \in S_g)\phi(s_t) + \beta^{loop-penalty}_\pi(s_{t+1}) \right], \tag{7}$$

where $\phi(s_t) := I(s_i \neq s_j, \forall \, 0 \leq i < t, t < j \leq T)$ is an indicator that judges whether there is a loop through $s_t$ in the trajectory $\tau$.

*Proof sketch*: The key idea is to convert Eq.6 to sum of the probability products of $p_\pi(\tau)$ and $I(s \in S_g)I(\tau^{no-loop})$, where $I(\tau^{no-loop})$ judges if there is not a loop in $\tau$. In addition, we mark the probability of reaching a state $s$ as $\rho_\pi(s) = P_\pi(s_0 = s) + P_\pi(s_1 = s, s_0 \neq s) \cdots$ and have:

$$p_\pi(\tau^{no-loop}_{-1} \in S_g) = \sum_{s_i} \rho_\pi(s_i) \sum_{t=i+1}^{T} I(s_t \in S_g)I(s_j \neq s_i, \forall \, i+1 < j < T). \tag{8}$$

We postpone the complete proof to the appendix. $\square$

So far we have derived that reducing the probability of loops can be achieved in sampling with multiplying $\phi(s_t)$ according to the signal of success or not in each collected trajectory, which is a method of online policy evaluation for state values.

### 4.2 ALGORITHM

In this subsection, we design three implementation methods by substituting the state-value function with LP into the loss functions of three commonly used RL algorithms, Monte Carlo (MC), Q-Learning (QL), and Proximal Policy Optimization (PPO) (Schulman et al., 2017). As discussed above, LP takes the form of multiplying the original state-value function with $\phi(s_t)$ as shown in

Fig. 4. Note that the indicator $\phi(s_t)$ can be implemented with many famous methods for measuring state similarity, such as GAN or VAE (Yu et al., 2019; Chen et al., 2016; Pathak et al., 2017). To that end, we derive three new adjusted loss functions, MC with Loop-Penalty (MC-LP), QL with Loop-Penalty (QL-LP), PPO with Loop-Penalty (PPO-LP) as follows:

$$\mathcal{L}_{MC-LP}(\pi_Q, \epsilon, s_t) \propto \mathbb{E}_{\tau \sim \pi_Q, \epsilon}\left[\|\sum_{k=t+1}^{T}\gamma^k r_k \phi(s_t) - Q(s_k, a_k)\|_2\right]_{\gamma=1}, \tag{9}$$

$$\mathcal{L}_{QL-LP}(\pi_Q, s_t) \propto \mathbb{E}_{\tau \sim \pi_Q}\left[\|(r_{t+1} + \gamma max_{a_{t+1}}Q(s_{t+1}, a_{t+1}))\phi(s_t) - Q(s_t, a_t)\|_2\right]_{\gamma=1}, \tag{10}$$

$$\mathcal{L}_{PPO-LP}(\pi, s_t) \propto -\mathbb{E}_{\tau \sim \pi, old}\left[min\left[A(s_t, a)\phi(s_t)\frac{\pi_k(a|s_t)}{\pi_{k,old}(a|s_t)}, clip\{\frac{\pi_k(a|s_t)}{\pi_{k,old}(a|s_t)}\}\right]\right], \tag{11}$$

where $\epsilon$ is the exploration rate of MC, $A(s_t, a)$ the advantage function and $clip\{\cdot\}$ the clipping function. Note that these algorithms all adopt online evaluation methods for value functions, because the probability of loops is related with the current policy. We choose QL-LP as representative to show our algorithm (Alg.1). The agent stores the state transitions collected in an episode into an online buffer $\mathcal{D}$ and use it to learn at the end of the episode. The loss function of LP-QL takes the product of $r_t + max_a Q(s_{t+1}, a)$ and $\phi(s_t)$ as the target Q-value $y_t$ in our algorithm.

## 5 EMPIRICAL RESULTS

In this section we aim to analyze the following three questions: 1) Does LP alleviate the uniformity of state values for success-rate optimization? 2) Does LP achieve better performance in terms of success rate, furthermore close to the highest possible success rate? 3) What is the difference between the policy with a high success rate and that with a high expected return?

### 5.1 TASK DESIGN

We design three environments to exhibit the problem and examine the effectiveness of our algorithm. 1) We use the aforementioned cliff-walking grid-world to show how our algorithm works in detail. 2) We construct a 3D first-person navigation task based on ViZDoom (Kempka et al., 2016) to examine whether LP is suitable for complex tasks. 3) We construct a robot (kinova jaco2) grasping task with CoppeliaSim (originally named V-REP) to examine the practicality.

In these three tasks, we constructed dangerous areas respectively, in which the agent fails with a certain probability: 1) windy area in the grid-world that makes the agent uncontrollably move down with a certain probability $p_{fall} = 0.1$ and fall down the cliff, 2) an area in the ViZDoom environment with a monster shooting at the agent, where the probability of failure depends on behaviors of the monster and the agent's random initial health, 3) a noisy area in the robot grasping task in which the arm is disturbed with a $0.2$ probability and may collide with the obstacle. These environments are illustrated in Fig. 5(a, b, c). The ViZDoom and robot grasping tasks only provide visual inputs for decision-making. To show our method is compatible with different RL algorithms, here we use three RL algorithms in three experiments: 1) QL and QL-LP in Grid-world, 2) MC and MC-LP in ViZdoom, 3) PPO and PPO-LP in Robot grasping. Other details are included in the appendix.

### 5.2 RESULTS ON CONVERGENCE AND SUCCESS RATE

First, we focus on the first question, i.e. whether our method alleviates the convergence problem of success-rate optimization. To reflect convergence, we plot curves about the change of success rate during training in Fig. 6, which is obtained by testing the policy ten times at the end of each training episode to calculate the success rates. It shows that there high variance when using MC ($\gamma = 1.0$) and QL ($\gamma = 1.0$) to optimize success rate, while our methods (marked by LP) can converge stably to a high success rate. These results indicate that: 1) the difficulty of convergence exists when optimizing success rate, 2) LP can stably optimize success rate.

Then, we try to answer the second question, i.e. whether our method achieves better performance than optimizing the expected discounted return. Furthermore, we check whether the success rate of our method can be close to 1. We test the model 1000 times at the end of training and calculated the success rate, as shown in Table.1. In our experiments, PPO-LP with $\gamma = 1.0$ has an obviously higher success rate than PPO optimized by expected discounted return with $\gamma = 0.7$ and that with

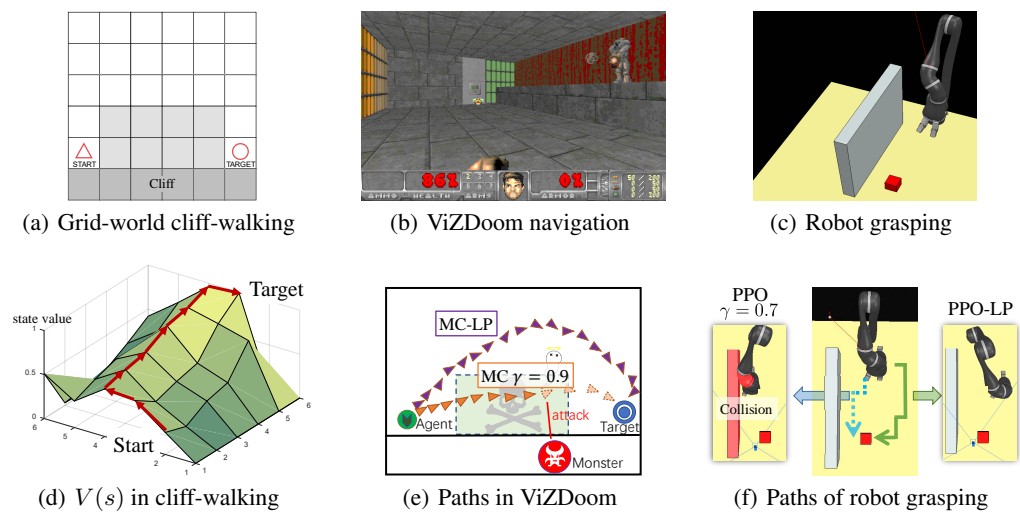

(a) Grid-world cliff-walking     (b) ViZDoom navigation     (c) Robot grasping

(d) $V(s)$ in cliff-walking     (e) Paths in ViZDoom     (f) Paths of robot grasping

Figure 5: Illustration of environments, value functions and policies

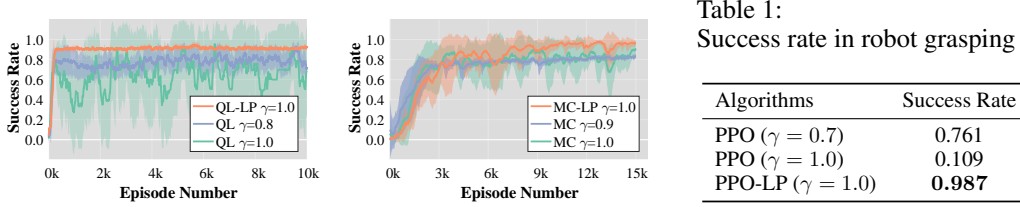

Table 1:
Success rate in robot grasping

| Algorithms | Success Rate |
|---|---|
| PPO ($\gamma = 0.7$) | 0.761 |
| PPO ($\gamma = 1.0$) | 0.109 |
| PPO-LP ($\gamma = 1.0$) | **0.987** |

Figure 6: Learning curve of cliff-walking (left), ViZDoom (right)

$\gamma = 1.0$, furthermore closer to 1. Results of success rate after training show that: 1) Optimizing with expected discounted return can not achieve the highest success rate in our experiments, and 2) the success rate of our method can be close to the highest.

## 5.3 VISUALIZATION OF STATE VALUES AND POLICIES

Lastly, we focus on our third question, i.e. what are the characteristics of policies trained with our method? We visualize the state values and the policy of our method in the grid-world task. As shown in Fig. 5 (d), there is no uniformity in state values and the trajectory bypasses the dangerous area. Then we visualize the policies of ours and policies got by optimizing expected discounted returns in ViZDoom and robot grasping, as shown in Fig. 5 (e,f). They show that the policies trained by maximizing success rate with LP tend to be reliable and risk-averse. On the contrary, the policies trained by maximizing expected discounted return tend to be risk-seeking.

## 6 DISCUSSION

This paper formally introduces the objective of success rate, analyzes the uniformity problem in directly optimizing success rate in RL, and proposes LP to alleviate it. As a potential impact, we think the discovery of the relationship between success rate and expected undiscounted return may imply that expected undiscounted return has some useful properties. As for future work, we hope to investigate different methods for measuring state similarity to improve the efficiency of LP. In addition, we think it is also beneficial to develop methods that alleviate the sparse-reward problem in optimizing success rate.

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

# A    APPENDIX

## A.1    PROOF OF THEOREM 1

In this section, we complete the proof of theorem 1.

**Theorem 1.** The success rate is a state-value function represented as an expected sum of undiscounted return, with the reward function $R(s)$ defined to take value of 1 if $s \in S_g$, 0 otherwise.

*Proof*: We segment the trajectories and generate sub-trajectories, $\tau \in \Gamma, \tau_{0:k} \in \hat{\Gamma}$ , where $k \in (0, T]$. Note that $\Gamma = \hat{\Gamma}$, because 1) $\forall \tau \in \Gamma$, we have $\tau_{0:T} \in \hat{\Gamma} = \tau, \Gamma \subseteq \hat{\Gamma}$, 2) $\tau_{0:k}$ is a trajectory, $\hat{\Gamma} \subseteq \Gamma$. Then the success rate $\beta_\pi(s_t)$ can be rewritten as the product sum of the probability of reaching $s_{t+k}$ and the indicator $I(\tau_{s_{t+k}} \in S_g)$ for all $s_{t+k}$:

$$\beta_\pi(s_t) = \sum_{k=1}^{T-t} \sum_{s_{t+k}} p_\pi(s_{t+k}|s_t) I(s_{t+k} \in S_g) \tag{12}$$

where $p_\pi(s_{t+k}|s_t)$ the probability of reaching $s_{t+k}$ from $s_t$. Then we substitute the formula of probability of reaching state $p_\pi(s_{t+k}|s_t) = \prod_{\hat{t}=t}^{t+k} \pi(a_{\hat{t}}|s_{\hat{t}})p(s_{\hat{t}+1}|s_{\hat{t}}, a_{\hat{t}})$ into $\beta_\pi(s_t)$:

$$
\begin{aligned}
\beta_\pi(s_t) =& \sum_{k=1}^{T-t} \sum_{s_{t+k}} \prod_{\hat{t}=t}^{t+k} \pi(a_{\hat{t}}|s_{\hat{t}})p(s_{\hat{t}+1}|s_{\hat{t}}, a_{\hat{t}})I(s_{t+k} \in S_g) \\
=& \sum_{a_t} \pi(a_t|s_t) \sum_{s_{t+1}} p(s_{t+1}|s_t, a_t)\{I(s_{t+1} \in S_g) \\
& + \sum_{a_{t+1}} \pi(a_{t+1}|s_{t+1}) \sum_{s_{t+2}} p(s_{t+2}|s_{t+1}, a_{t+1})\big[\sum_\tau I(\tau_{-1} \in S_g)p(\tau|s_{t+1})\big]\} \\
=& \sum_{a_t} \pi(a_t|s_t) \sum_{s_{t+1}} p(s_{t+1}|s_t, a_t)\left[I(s_{t+1} \in S_g) + \beta_\pi(s_{t+1})\right]
\end{aligned}
\tag{13}
$$

If we compare success rate of states $\beta_\pi(s_t)$ with the state-value function $V_\pi(s_t) = \sum_{a_t} \pi(a_t|s_t) \sum_{s_{t+1}} p(s_{t+1}|s_t, a_t)\left[r_{t+1} + V_\pi(s_{t+1})\right]$, we will find that the success rate is a kind of undisounsted return with $\{0,1\}$-binary reward of $I(s_t \in S_g)$, which completes the proof.

## A.2 PROOF OF THEOREM 3

In this section, we complete the proof of theorem 3.

**Theorem 3.** The state-value function policy for Eq.6 is

$$
\beta_\pi^{loop-penalty}(s_t) = \mathbb{E}_{\tau\sim\pi}\left[I(s_{t+1} \in S_g)\phi(s_t) + \beta_\pi^{loop-penalty}(s_{t+1})\right],
\tag{14}
$$

where $\phi(s_t) := I(s_i \neq s_j, \forall\, 0 \leq i < t, t < j \leq T)$ is an indicator that judges whether there is a loop through $s_t$ in the trajectory $\tau$.

*Proof*: The key idea is to convert Eq.6 to sum of the probability products of $p_\pi(\tau)$ and $I(s \in S_g)I(\tau^{no-loop})$, where $I(\tau^{no-loop})$ judges if there is not a loop in $\tau$. In addition, we mark the probability of reaching a state $s$ as $\rho_\pi(s) = P_\pi(s_0 = s) + P_\pi(s_1 = s, s_0 \neq s)\cdots$ and have:

$$
\begin{aligned}
p_\pi(\tau_{-1}^{no-loop} \in S_g) =& \sum_{s_i}\rho_\pi(s_i) \sum_{t=i+1}^{T} I(s_t \in S_g)I(s_j \neq s_i, \forall\, i+1 < j < T) \\
=& \sum_{s_i}\rho_\pi(s_i)\big[\sum_{a_{i+1}} \pi(a_{i+1}|s_{i+1}) \sum_{s_{i+1}} p(s_{i+2}|s_{i+1}, a_{i+1})\big[I(s_{i+2} \in S_g)I(s_j \neq s_i, \forall\, i < j < T)+ \\
& \sum_{a_{i+2}} \pi(a_{i+2}|s_{i+2}) \sum_{s_{i+2}} p(s_{i+3}|s_{i+2}, a_{i+2})\big[\sum_{t=i+3}^{T} I(s_t \in S_g)I(s_j \neq s_i, \forall\, i+1 < j < T)\big]\big]\big] \\
=& \sum_{s_i}\rho_\pi(s_i)\mathbb{E}_{\tau\sim\pi}\big[I(s_{i+1} \in S_g)I(s_j \neq s_i, \forall\, i+1 < j < T) \\
& \mathbb{E}_{\tau\sim\pi}\big[\sum_{t=i+2}^{T} I(s_t \in S_g)I(s_j \neq s_i, \forall\, i+2 < j < T)\big]\big] \\
=& \mathbb{E}_{\tau\sim\pi}\big[I(s_t \in S_g)I(s_j \neq s_i, \forall\, 0 < i < t, t < j \leq T)+ \\
& \mathbb{E}_{\tau\sim\pi}\big[\sum_{k=t+1}^{T} I(s_k \in S_g)I(s_j \neq s_i, \forall\, 0 < i < k, k < j \leq T)\big]\big]
\end{aligned}
\tag{15}
$$

Then let $\phi(s_t) = I(s_j \neq s_i, \forall\, 0 < i < t, t < j \leq T)$, and we have

$$
p_\pi(\tau_{-1}^{no-loop} \in S_g) = \mathbb{E}_{\tau\sim\pi}\big[I(s_{t+1} \in S_g)\phi(s_t) + \mathbb{E}_{\tau\sim\pi}\big[\sum_{t+2}^{T} I(s_{t+2} \in S_g)\phi(s_{t+1})\big]\big]
\tag{16}
$$

Considering $\beta_\pi^{loop-penalty}(s_t) = \mathbb{E}_{\tau\sim\pi}\big[\sum_t^T I(s_{t+1} \in S_g)\phi(s_t)\big]$, the $\beta_\pi^{loop-penalty}(s_t)$ can also be written as a recursive form:

$$
\beta_\pi^{loop-penalty}(s_t) = \mathbb{E}_{\tau\sim\pi}\big[I(s_{t+1} \in S_g)\phi(s_t) + \beta_\pi^{loop-penalty}(s_{t+1})\big]
\tag{17}
$$

which completes the proof.

Table 2: Learning rate

| Algorithms | Learning Rate |
|------------|---------------|
| QL | 0.1 |
| MC | 0.01 |
| PPO | 0.0001 |

Table 3: Exploration rate

| Algorithms | Exploration Rate |
|------------|------------------|
| QL | 0.3 |
| MC | random choice from [0,0.5] |
| PPO | 0.0002 (coefficient) |

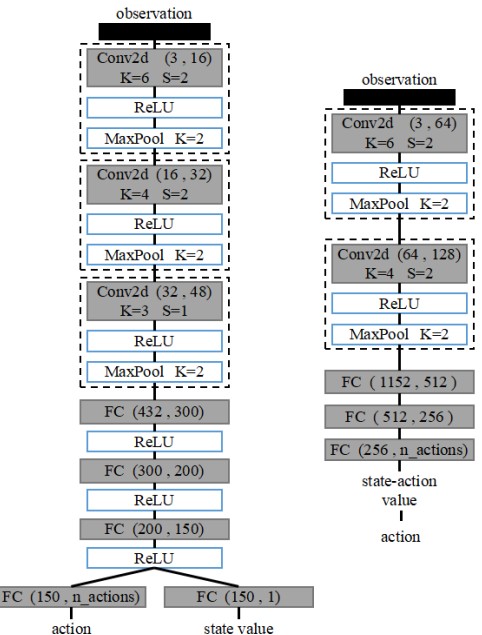

Figure 7: Schematics of the networks of 1) PPO the left 2) MC the right

## A.3 EXPERIMENT DETAILS

We set the same hyper-parameters for models of our algorithms and the baselines. The learning rates and exploration rates are set as the Table 2 and Table 3, in which the exploration rate is expressed with the coefficient of policy entropy. The parameter for gradient clipping in PPO $\epsilon_{clip}$ is set to 0.1. We implement $\phi(s_t)$ with environmental information (position information in ViZDoom and pose information of arm in robot grasping) as signals in the training, but not use the information in the testing. The models we used in Grid-World tasks are tabulars which has the same shapes as the environments' and those in ViZDoom tasks and Robot tasks are neural-networks. The models of networks are shown in Fig. 7. And the inputs are shown in Fig. 8.

## A.4 NUMERICAL RESULTS ON STATE-ACTION VALUES

We show the state-action values of MC-LP ($\gamma = 1.0$), QL ($\gamma = 0.6$) and QL ($\gamma = 1.0$) after training as Fig. 9. State-action values of QL ($\gamma = 1.0$) show the phenomenon of uniformity. State-action values of QL ($\gamma = 0.6$) has no uniformity but tend to be risk-seeking. State-action values of MC-LP ($\gamma = 1.0$) has no uniformity and perform conservative, with high success-rate.

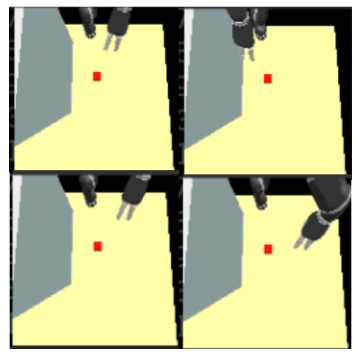 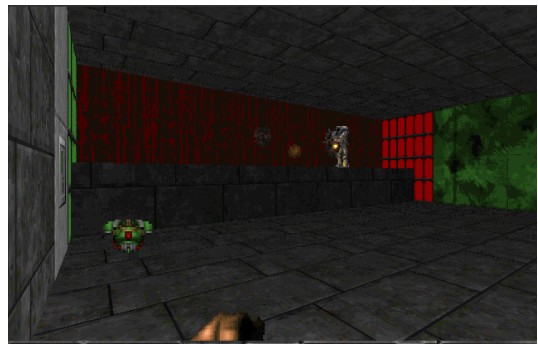

(a) Observations in robot grasping      (b) An observation in ViZDoom

Figure 8: Exhibition of the inputs of networks

(a) State-action values of QL ($\gamma = 0.6$)      (b) State-action values of QL ($\gamma = 1.0$)

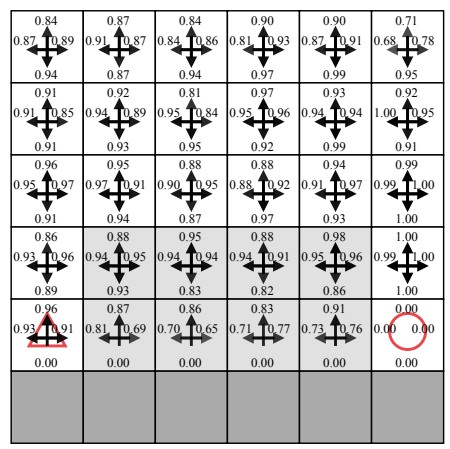

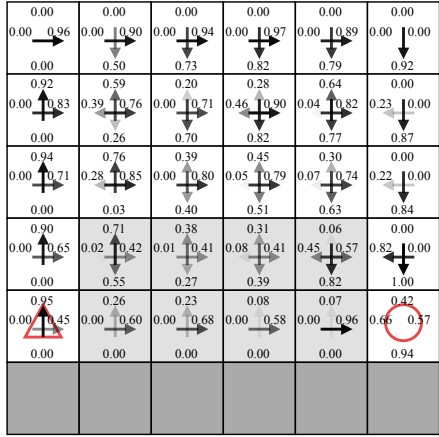

(c) State-action values of MC-LP ($\gamma = 1.0$)

Figure 9: Illustration of state-action values

