# OpenReview forum: "Success-Rate Targeted Reinforcement Learning by Disorientation Penalty"
_ICLR.cc/2021/Conference — Reject_

### Official Review · AnonReviewer2 · 2020-10-18
**Theoretical analysis partly known, partly erroneous. Algorithmic contribution needs comparison to simpler alternatives and more extensive evaluation.**

**Rating:** 2
**Confidence:** 5

**Review:**

------------------------------------------
POST-REBUTTAL COMMENTS

Thanks for your comments.

Re: A2, time augmentation in finite-horizon settings increases the size of the state space you need to keep in memory by at most a factor of 2... But in any case, further discussion of this issue will have to wait until the revision with the additional experiments that you mentioned.

Re: A5, regarding "D" being analogous to "gamma" -- fair enough, but this approach is still a meaningful baseline to compare against.

Regarding the statement "contrary to this, initializing state values pessimistically requires task-specific knowledge", this isn't accurate. In a MAXPROB instance, initializing the the value function at all states to 0 is a problem-independent pessimistic initialization. Of course, task-specific knowledge may help design a better one, but it isn't _necessary_.

In conclusion, as the original review mentioned, I believe that the presented "loop penalty" idea may well have conceptual merit, but I encourage you to think more carefully how you "sell" it, because so far neither the original submission nor the rebuttal present convincing arguments that it is better than the alternatives either theoretically or empirically.

---------------------------------------------------


The paper argues that in many RL settings the expect discounted reward criterion common in RL is less appropriate than undiscounted success rate maximization, which this paper claims to introduce. The paper argues that the two result in different solutions, points out that success rate maximization can result in instability of existing RL approaches due to introducing "uniformity" of state values within state loops, and proposes modified losses for PPO, Monte-Carlo, and Q-learning algorithms that allows them to optimize the success rate more reliably than they are able to using their standard losses.

The paper's motivation is sound: the discounted-reward criterion is indeed conceptually less appropriate than success rate maximization for goal-directed decision-making problems. Unfortunately, however, the paper's claimed contributions in addressing this issue are not novel, and largely flawed:


1) Contrary to the paper's claimed contribution, success rate maximization in MDPs isn't new. Paper [1] introduced this criterion, calling it "MAXPROB", and analyzed it mathematically. Without an official name this criterion was known even before that (see, e.g., [1a]). The analysis in [1] focuses on MDP planning, i.e., assumes known model, but the mathematical properties pointed out there that affect value iteration convergence on these MDPs hold in the RL setting just the same.

2) The submission's claims about the success rate/MAXPROB criterion causing the instability of value iteration based approaches are partly imprecise and partly outright mistaken. In the intro, the paper states, "this expression belongs to undiscounted problems and the convergence of value iteration
often cannot be guaranteed (Xu et al., 2018)". First of all, I couldn't find any such claims in that paper. Second, that paper doesn't deal with *finite-horizon* MDPs, whereas this submission does, and the problem is that in finite-horizon MDPs succcess rate maximization poses no issues for value iteration at all. The reason is that value iteration in finite-horizon MDPs such as those in Section 3.1 essentially operates on time-augmented state space, needs a single backward pass from states with 0 steps-to-go to states with T steps-to-go in order to compute the optimal value function, and its convergence doesn't depend on the properties of the reward function.
Note that Section 3.2.C and Figure 3 that it refers to doesn't talk about value iteration's convergence difficulties during success rate maximization in finite-horizon MDPs from Section 3.1 anymore. It just states in a hand-wavy way that there are some difficulties with this criterion, but doesn't explain exactly what they are.

3) The concept of uniformity, as flawed as it is for finite-horizon MDPs defined in Section 3.1, is actually not novel and is subsumed by previously published analysis, again from paper [1]. [1] analyzes the success rate/MAXPROB criterion in *infinite-horizon* undiscounted MDPs with absorbing non-goal states, where vanilla value iteration truly has difficulties with this criterion, and shows that these difficulties are indeed caused by state values being uniform in loops/strongly connected components of the MDP's transition graph. In particular, as shown there, this uniformity introduces additional fixed points of the Bellman backup operator that value iteration relies on, and value iteration can converge to any of these fixed points as a result.

4) Several of the submission's other theoretical claims are quite sloppy as well. For instance, in the intro there is a statement "We believe that success rate is different from expected discounted return". I think this notion of difference should be made sufficiently precise so as to take faith out of the equation. Same goes for the loop penalty surrogate criterion in Section 4.1. Does optimizing it, at least in tabular MDPs using vanilla value iteration, result in obtaining a policy that maximizes success rate?

5) By itself, the loop penalty criterion looks new. However, a) as mentioned above, it's not clear whether it is a heuristic or has actual optimality guarantees w.r.t. success rate optimization and b) more importantly, papers [1] and [2] suggest at least two alternatives to fixing value iteration's convergence for success rate maximization (in infinite-horizon undiscounted MDPs):

   (a) As shown there, for value iteration/Q-learning-like methods, initializing state values *inadmissibly* (i.e., pessimistically in the face of uncertainty) and amending the Bellman backup operator to deal with "value uniformity" yields an optimal algorithm for this criterion.

   (b) Turn the success rate maximization MDP into a stochastic shortest path MDP (see [3] or almost any textbook by Bertsekas and Tsitsiklis) by assigning a positive cost to every action, introducing a "cap" D on the highest possible state cost, and minizing the undiscounted expected cost (see stochastic shortest path MDPs with finite dead-end penalty in [2]). D is a hyperparameter, and for *any* such cost assignment there exists a D s.t. the optimal policy for this surrogate MDP will that maximizes success rate in the original MDP. Results from [4] may even help prove something about convergence rate of this approach, although empirically convergence speed and resulting policy (note that there are generally many success rate-maximizing policies) will depend on the specific cost function choice.

At least method (b) is conceptually simpler than the loop disorientation penalty, and may even be theoretically similar to the latter, and provides a natural baseline for the proposed approach.

6) Last but not least, the empirical evaluation is too limited to be able to assess the merits of the proposed approach. While there are certainly problems where optimizing for success rate directly is preferable to optimizing the discounted reward, the use of discount factor in RL is important in many problems for mitigate the effects of estimation errors -- see, e.g., Xu et al. 2018 and [5]. Therefore, to get a better picture of whether success rate optimization is worth it in practice, one would need a more extensive evaluation on benchmarks such as goal-directed Atari or Procgen games and/or more complex robotics scenarios.


Thus, despite studying an interesting topic, I think this work needs to be significantly revised and extended before publication.



[1] Kolobov, Mausam, Weld, Geffner. "Heuristic search for generalized stochastic shortest path MDPs" ICAPS-2011

[1a] Little, Thiebaux "Probabilistic Planning vs Replanning" An ICAPS-2007  workshop

[2] Kolobov, Mausam, Weld. "A Theory of Goal-Oriented MDPs with Dead Ends" UAI-2012

[3] Bertsekas, Tsitsiklis. "An Analysis of Stochastic Shortest Path Problems" Mathematics of Operations Research, 1991

[4] Yu, Bertsekas. "On Boundedness of Q-Learning Iterates for Stochastic Shortest Path Problems" Mathematics of Operations Research, 2013

[5] Jiang, Kulesza, Singh, Lewis. "The Dependence of Effective Planning Horizon on Model Accuracy" AAMAS-2015

---

> ### Author Response · Authors · 2020-11-25
> **Response to Reviewer 4**
>
> Thank you very much for your careful reading and detailed comments. Your review is precious and important for our future work.
>
> A1: Thanks for your introduction to the paper [1]. We realize that our contribution 1 is an overstatement and we will revise it in a future version.
>
> A2: Augmenting time t to state is a possible solution for alleviating “uniformity” in finite horizon settings. However, this augmentation enlarges the state space and makes training less efficient. In this paper, we would like to provide a more efficient method that does not rely on time-augmentation.  We performed an empirical comparison of Loop-penalty Q-learning and time-augmented Q-learning, in which our method can achieve faster convergence, and we will add this comparison in our revision.
>
> A3 & A4 We will revise our analysis in a future version.
>
> A5: Our Loop-penalty method is based on our heuristic that invalidating loops in sampled trajectories can avoid the “uniformity” problem, and our method is general and applicable to most RL problems and algorithms. Contrary to this, initializing state values pessimistically requires task-specific knowledge.
>
> As for turning the success rate maximization MDP into a stochastic shortest path MDP, we agree that it is a solution for fixing VI, and we also want to mention that it relies on task-specific knowledge to design suitable D. From this perspective, we feel that this method is analogous to tuning the discount factor: they can both eliminate the problem, but there is no such a single setting that suits all tasks. Instead, our method provides an alternative that does not require such prior knowledge.
>
> A6: We will consider your suggestions in our future work. Thank you.

---

### Official Review · AnonReviewer3 · 2020-10-28
**Creative and relevant idea, but missing crucial detail.**

**Rating:** 3
**Confidence:** 4

**Review:**

The authors propose a new set-up for reinforcement learning which considers undiscounted episodic returns and introduces a loop-penalty to ensure that all episodes terminate and that the returns are bounded. In the tabular case, the loop-penalty zeros out the reward if a loop is detected in the current episode. For continuous state-spaces, the authors propose to detect loops using methods that estimate state-similarity.

The proposed formulation is relevant as well as novel and creative. While discount factors have been questioned repeatedly in recent years, research looking into alternative formulations remains sparse. The most common formulation for non-discounted reinforcement learning, average-reward RL, suffers from the difficulty of estimating the average reward effectively and alternative formulations could be beneficial; however, the paper is lacking in details when it comes to its experimental set-up as well as the method used for non-tabular methods and the analysis is often trivial.

I would encourage the authors to spend more time on the practical questions of how to estimate the loop-penalty in non-tabular domains. The positive results on vizdoom as well as a robotics task imply that detecting such loops is possible, but there is no discussion of how this is done, in which cases it can work well and in which cases it is still too difficult. It would also be good if the empirical evaluation would include higher, more realistic discount factors such as 0.9 for robot grasping or 0.99 for VizDoom.

With regards the analysis, I believe that the following is problematic:
* A large part of the analysis focuses on uniformity. The authors first show that the value-function has no unique solution on ergodic states with zero reward and then draw a connection between this and the phenomenon illustrated in Figure 2. My first issue here is that the result is well-known in the average reward scenario of which this is a special case. The claim immediately follows and the proof in the paper is unnecessary. My second issue is that it is unclear how the constant offset that can be applied to the value function relates to the flatness of the value function shown in Figure 2. The constant offset can be applied to non-discounted value functions even when the value function does not exhibit this issue.
* Theorem 3 writes out the value function for RL with loop-penalty. The loop-penalty makes the reward non-markovian, therefore it is unclear what theoretical insights can be gained from this theorem.
* Theorem 1 is proving that the episodic return corresponds to success probability if the reward is 1 for success and 0 otherwise. This is trivial and the proof is overly complicated. I don’t believe a proof for this is necessary.

---

> ### Author Response · Authors · 2020-11-25
> **Response to Reviewer 3**
>
> Thank you very much for your thoughtful comments.
>
> A1: We agree that the constant offset alone does not lead to the flatness of the value function. However, when S_e exists, the state values are the same among these states, while values for the surrounding states may be different from states in S_e. If state values among S_e are higher than the surroundings, it will result in a sub-optimal policy from which traditional value iteration cannot escape.
>
> A2: The primary purpose of theorem 3 is to show that Loop-penalty can be realized by multiplying Phi to certain parts of trajectories. Although the value function is non-Markovian, the resulting algorithm is practically feasible. Our experiments further show that Loop-penalty is beneficial for improving training stability and the final performance.
>
> A3: We will carefully consider your suggestion about relevant proofs in existing publications and revise ours in the future work. Thank you.

---

### Official Review · AnonReviewer4 · 2020-10-28
**Interesting idea that could develop into a good paper**

**Rating:** 4
**Confidence:** 4

**Review:**

### Paper Summary

This paper proposes an alternative surrogate objective for optimizing success rate in episodic tasks with bounded time horizon. Rather than optimize a discounted 0-1 loss (say with discount factor 0.99), the authors suggest to optimize the undiscounted 0-1 loss where reward is counted only for trajectories that do not contain loops. They call this a loop penalty, and show that it can work in 3 appropriate environments.

### Review Summary

I like the idea and think it could develop into a good paper. That said, I have a couple technical issues (they should be easy enough to fix), take minor issue with the current presentation (again, this should be easy to fix), and importantly, would want to see this method tested against the the right baselines (easy to fix, but require some work) before I can recommend acceptance.

### Pros

I like the overall approach and idea of introducing an alternative surrogate task for purposes of optimizing the success rate. If you can show either (1) that this actually leads to easier optimization in practice than the alternative surrogates (discounted reward specification, and also direct optimization (see below)) or (2) that provides certain other advantages to the alternative surrogates, this would make for a good contribution.

### Cons

The framing here is a little off. You might consider the framing from some other papers (e.g., [1], [2], [3]) that introduce approximate task specifications as a surrogate training objective. My reasoning here is that (1) I think the idea of optimizing success rate is well recognized in the literature, and (2) once you introduce the loop penalty you are no longer optimizing undiscounted return / success rate, but using a surrogate task specification (more on this below). The appropriate baselines here are (1) discounted reward with a properly chosen discount, and (2) direct optimization of the success rate (see below).

The first contribution (introduction of success rate and association of success rate with undiscounted return) is obvious, and could be presented in a much more compact form.  Actually, as presented, I believe the statement and proof of Theorem 1 are incorrect given that in the definition of “success rate” the authors require the agent to be successful with at most T steps. The proof sketch is not a proper sketch, but only the first step. The full proof in the Appendix drops the reference to T at the second step, and the final step (identifying it with the binary undiscounted return) ignored that T < infinity. This might be fixed if the states themselves include the current timestep “t”, but then Figure 2 wouldn’t look like Figure 2 (we would have a different shape for each time step), we would haven’t plauteaux, and the Theorem 2 would be irrelevant.

I think Equation (7) and Theorem 3 are incorrectly stated. You should be multiplying both the reward and target by phi(s_t), as you are doing, e.g., in Eqs (9), (10). Glancing at the proof, Equation (16) seems correct, so it is just the last “expressing in recursive form” step that is incorrect.

In stochastic environments, Loop Penalty can introduce the same kind of approximation as gamma < 1. E.g., it will favor a policy that produces very long non-loopy trajectories over one that produces short trajectories that are more likely to have loops. So this is actually very similar to using gamma < 1 as the surrogate objective.  I.e., it is a *surrogate* for the true success rate.  So, in your first experiment, for example, you should have a gamma < 1 baseline, e.g., gamma = 0.99. Like the loop penalty, it is also a surrogate objective. In the second experiment, why are you using gamma = 0.7 as the baseline? This discount factor seems unreasonably small, so of course it does poorly. As a rule of thumb, you can use a value for which the effective horizon (1 / (1-gamma)) is approximately the expected trajectory length. E.g., 0.98 for T = 50.

As long as we are allowing our backup equation to consider the entire trajectory. Why not just directly optimize the success rate: i.e., what this episode successful or not? This seems like it would be almost the same from an implementation perspective and would make for a good additional baseline.

### References

Here are a few recent papers that modify the task specification / Bellman equation in order to improve the optimization:

- [1] Van Seijen, Harm, Mehdi Fatemi, and Arash Tavakoli. "Using a logarithmic mapping to enable lower discount factors in reinforcement learning." Advances in Neural Information Processing Systems. 2019.

- [2] De Asis, Kristopher, et al. "Fixed-horizon temporal difference methods for stable reinforcement learning." arXiv preprint arXiv:1909.03906 (2019).

- [3] Tessler, Chen, and Shie Mannor. "Maximizing the Total Reward via Reward Tweaking." arXiv preprint arXiv:2002.03327 (2020).

---

> ### Author Response · Authors · 2020-11-25
> **Response to Reviewer 2**
>
> Thank you very much for your comments. Since the comments were not organized by entries, we provide explanations based on our summary of your points.
>
> A1: We feel that we should have made clearer the finite time-horizon in our analysis. Our current analysis is based on the thought that most reinforcement learning methods are using bootstrapping without the knowledge of the remaining time, thus we did not include time-horizon T. As for methods that do not use bootstrapping, such as Monte Carlo, we provide empirical results in the experiment section.
>
> A2: We acknowledge that augmenting time t into states may be a solution that fixes that “uniformity” problem. However, this will severely enlarge the state space and reduce the efficiency of training. In contrast, our method can stabilize training without augmenting the state space. We performed an empirical comparison of Loop-penalty Q-learning and time-augmented Q-learning, in which our method can achieve faster convergence, and we will add this comparison in our revision.
>
> A3: In our second experiment, we used gamma=0.7 to show that the task cannot be well-solved with a low discount rate. As discussed in the introduction part, when gamma is close enough to 1, it can serve as a correct surrogate for success rate. However, the selection of gamma depends on prior knowledge of the task and may introduce difficulties in training stability. We agree that some prior knowledge (such as the example of gamma=0.98 for T=50, as you mentioned) may work in many cases, however, there is still no guarantee of its correctness.
>
> A4: In our experiments, the results gained with gamma=1.0 without application of Loop-penalty are the baselines of directly optimizing success rate. Concretely they are labeled as QL-gamma=1.0, MC-gamma=1.0, and PPO-gamma=1.0. The results show that the corresponding training processes are not stable and do not attain the optimal success rate.

---

### Official Review · AnonReviewer1 · 2020-10-29

**Rating:** 4
**Confidence:** 4

**Review:**

This paper explores optimizing a binary undiscounted success measure in RL, identifies the issue with learning in such undiscounted environments where many states have the same value (because the policy has no urgency to do anything), propose a “loop penalty” to fix the issue, and show that the loop penalty enables learning in this setting. The problem that the paper identifies of learning with a binary success measure without discounting is an interesting one, since discounting is a somewhat “hacky” solution and it results in suboptimal behaviors in practice. The presented proofs have some nice arguments why the value issue happens in this setting. The results are positive, although not outstanding, showing modest gains in three domains: grid world, VizDoom, and robot grasping. However, there are some issues with correctness and coverage of related work that I discuss below.

Related work - VICE (Ghosh et al. 2018) is cited but not treated fairly. The “AT” or “ANY” queries seem to consider exactly the issue discussed in this paper and do not require discounting. The solution presented in that paper, which modifies the bootstrapping update, seems a more general solution than the loop penalty presented in this paper.

Termination: the termination condition described is to end the episode on reaching a goal state or on reaching a maximum T. First, it seems that practically for policy optimization with function approximation, if termination is handled correctly in bootstrapping (ie. do not bootstrap for the target value in transitions that are terminal) the maximum T will actually have a similar effect as discounting, which perhaps explains why in Fig 6 the base RL methods still learn with discount = 1. Second, it seems that this finite horizon is not discussed in the proofs - it’s not clear to me that the same issue exists even in tabular MDPs if you use finite-horizon value iteration.
Page 4: V = R + gamma P V, not P R + gamma P V. Then in the same paragraph: what if there are multiple subsets S_e of the state that form an irreducible Markov chain, or there is no such subset with more than 1 element, or the entire MDP is irreducible? (I don’t think this breaks the proof, it should just be clarified “which” subset you choose because currently its ambiguous)

Theorem 3: the indicator phi(s_t) seems like it should be a function of tau as well

“Note that the indicator φ(st) can be implemented with many famous methods for measuring state similarity” - I did not understand this, do you threshold the difference between states in a latent space to get the resulting indicator? If so, how do you choose the threshold, it would seem to be highly dependent on the environment dynamics.

---

> ### Author Response · Authors · 2020-11-25
> **Response to Reviewer 1**
>
> Thank you very much for your comments. Below we provide our explanations for the problems.
>
> Q1: Related work - VICE (Ghosh et al. 2018) is cited but not treated fairly.
> A1: We believe that we did not cite the paper you mentioned. In our paper, “Ghosh et al. 2018” refers to “Divide-and-conquer reinforcement learning”. We will add VICE in our revision.
>
> Q2: (1) If termination is handled correctly in bootstrapping, the maximum T will have a similar effect as discounting.
> A2: (1) We acknowledge that termination should be handled carefully and that not bootstrapping at terminal states is to some extent beneficial for learning. However, this does not eliminate the “uniformity” problem we discussed in our paper. We did not bootstrap for terminal states in our experiments, and we did observe an obvious performance gap between undiscounted and discounted settings. From our experiments, it is discounting or introducing our loop penalty that effectively improves the performance.
>
> Q2: (2) The finite horizon is not discussed in the proofs.
> A2: (2) Thanks for pointing this out. We agree that it would be better if we can add more details about the time-horizon in our proof. The current version of the proof is based on our thought that most reinforcement learning methods use bootstrapping without the knowledge of the remaining time. Thus, a derivation without T has a larger range of applications. As for methods without bootstrapping, such as Monte Carlo, we provide empirical results in Section 5.
>
> Q2: (3) Page 4: V = R + gamma P V, not P R + gamma P V
> A2: (3) We believe that there should be a transition probability matrix before the reward vector in our formulation. The reason is that in our model the immediate reward depends on the next state instead of the current state. With P before R, we correctly represent the Bellman equation. We agree that there are different forms of the Bellman equation, but we want to adopt the one introduced in Richard S. Sutton’s “Reinforcement Learning: An Introduction” so that it is easier for readers to understand.
>
> Q2: (4) What if there are special cases of S_e or the entire MDP?
> A2: (4) If there are multiple subsets of S_e, each of them will exhibit the “uniformity”, and as we analyzed in section 3.2.C, there may be problems with training in each of the subsets. If there are no qualified S_e, we should not be seeing the “uniformity” problem. We think that the entire MDP cannot be S_e because we assume that there is at least one goal state that is absorbing.
>
> Q3: There is a problem with phi(s_t).
> A3: We will revise them in a future version of the paper.
>
> Q4: How to threshold the difference between states?
> A4: We agree that for some tasks with high-dimensional state spaces there will be some difficulties in determining the suitable threshold. We feel that this is an interesting and important problem and will include it in our future work.

---

> > ### Comment · AnonReviewer1 · 2020-11-25
> > **Response to rebuttal**
> >
> > Thank you for the response.
> >
> > (1) Alright, the other paper seems quite relevant to this work.
> >
> > (3) OK, understood.
> >
> > (4) S_e: makes intuitive sense, I think the proof needs to handle these cases more carefully though.
> >
> > Q4: Agreed that it is interesting future work to pick a similarity metric, but also the current paper did not describe in enough detail what was actually done in the experiments.

---

### Decision · Program_Chairs · 2021-01-07
**Final Decision**

**Decision:**

Reject

**Comment:**

Despite the fact that some of the reviewers found the idea interesting, none of them believe that the paper is ready to be published at this stage. For example, better comparison with existing/similar work, and more solid argument on why the idea is better than alternatives are mentioned. All considered, unfortunately I cannot recommend acceptance of this paper in its current form. I encourage the authors to consider these comments and revise their paper accordingly.